# Changes in Polyphenolics during Storage of Products Prepared with Freeze-Dried Wild Blueberry Powder

**DOI:** 10.3390/foods9040466

**Published:** 2020-04-09

**Authors:** Laura Lavefve, Cindi Brownmiller, Luke Howard, Donovon Reeves, Sean H. Adams, Jin-Ran Chen, Eva C. Diaz, Andy Mauromoustakos

**Affiliations:** 1Department of Food Science, University of Arkansas, 2650 North Young Avenue, Fayetteville, AR 72704, USA; ldlavefv@uark.edu (L.L.); cbrownm@uark.edu (C.B.); dxr017@uark.edu (D.R.); 2Department of Pediatrics, University of Arkansas for Medical Sciences, Little Rock, AR 72202, USA; shadams@uams.edu (S.H.A.); chenjinran@uams.edu (J.-R.C.); ECDiazfuentes@uams.edu (E.C.D.); 3Arkansas Children’s Nutrition Center, 15 Children’s Way, Little Rock, AR 72202, USA; 4Arkansas Children’s Research Institute, 1 Children’s Way, Little Rock, AR 72202, USA; 5Agricultural Statistics Lab, 104 Agricultural Annex, University of Arkansas, Fayetteville, AR 72701, USA; amavro@uark.edu

**Keywords:** anthocyanin, chlorogenic acid, flavonol, polymeric color, storage, wild blueberry

## Abstract

Wild blueberry (WBB) powder can be added to the formulation of foods to encourage consumption of health-promoting polyphenolics, but the stability of polyphenolics throughout storage is important. We determined the stability of polyphenolics in five products (ice pop, oatmeal bar, graham cracker cookie, juice, and gummy product) prepared with WBB powder. Samples stored at 21 °C, 4.4 °C, or −20 °C (ice pops only) were analyzed at 0, 2, 4, 6, and 8 weeks for polyphenolic content and percent polymeric color. Total anthocyanins decreased over storage and storage temperatures in all products. However, the ice pop and the refrigerated juice both retained over 90% of their initial total anthocyanin content. The refrigerated oatmeal bar also showed good retention of anthocyanins (86%), but the gummy product retained only 43% and 51% when stored at 4.4 °C or 21 °C, respectively. The lower amount of polyphenolic compounds recovered in the gummies stored at 4.4 °C compared to 21 °C may be attributed to reduced extraction efficiency as a result of gel hardening at refrigerated temperature. Chlorogenic acid and flavonols were generally more stable than anthocyanins throughout storage.

## 1. Introduction

Lowbush “wild” blueberries (*Vaccinium angustifolium* Ait) are considered a nutrient-rich healthy food, due in large part to their exceptional phenolic content [1,2] and antioxidant activity [2]. Lowbush blueberries are particularly rich in anthocyanins and the anthocyanin profile is complex compared with other fruits [3,4]. They contain five of the six anthocyanidins commonly found in nature (delphinidin, cyanidin, petunidin, peonidin, and malvidin), which can have three different sugar moieties attached (galactose, glucose, and arabinose) as well as acyl groups such as acetyl-, malonyl-, or coumaryl- also attached to the sugar moieties [4,5]. Blueberries are also rich in proanthocyanidins [1,6], chlorogenic acid [7,8], and flavonols [7,9].

Diets rich in blueberries or their polyphenolic-rich extracts have been associated with lower cardiovascular risk, weight gain and metabolic syndrome, and neurological diseases (reviewed in [10]). In addition, studies involving blueberries have identified polyphenolic-derived phenolic acids that improve cell differentiation and proliferation of osteoblasts in vitro and promote bone growth and limit bone loss in rodents [11,12,13]. These health-promoting effects are due to a myriad of mechanisms associated with blueberry polyphenolics, including prevention of oxidative stress and inflammation, and vaso- and lipid modulation [14,15,16]. Many human studies reporting positive health outcomes have used freeze-dried wild blueberry (WBB) powder [17,18,19,20,21], which is a natural source of concentrated polyphenolics. However, the freeze-dried WBB powder may be tart or astringent and not always palatable to consume. This can be problematic in feeding trials in children and adults. In our previous work, we developed five food products (gummy, oatmeal bar, graham cracker cookie, juice, and ice pop) prepared with freeze-dried WBB powder that were evaluated for children’s acceptability and desire to eat [22]. These results are useful in designing food products as well as menu items that could be used in clinical trials of WBB-rich diets. In addition to evaluating sensory properties, it is important to validate the storage stability of polyphenolics in these products, before use in clinical trials, to ensure that a consistent dose of polyphenolics can be maintained. Blueberry polyphenolics, especially anthocyanins, are unstable in various processed forms such as juices, jams, purees, and canned berries when stored at ambient temperature [23,24,25]. Additionally, anthocyanins in freeze-dried WBB powder are susceptible to degradation when stored at ambient temperature with a reported half-life of 139 days at 25 °C [26].

The mechanism responsible for loss of anthocyanins during storage is unknown, but anthocyanin losses are commonly accompanied by increased polymeric color values, suggesting that anthocyanins form polymers with proanthocyanidins [27]. In addition to polymerization, many other factors can affect the stability of anthocyanins including exposure to elevated temperatures, light, oxygen, metals, sugars, and ascorbic acid [28]. At present, refrigeration of blueberry products such as jam [24] and juices [29,30] is the best approach to mitigate polyphenolic losses during storage.

This study was undertaken to determine the stability of anthocyanins, flavonols, chlorogenic acid, and percent polymeric color in five blueberry products prepared with freeze-dried WBB powder. Gummy, oatmeal bar, graham cracker cookie, and juice were stored at 21 °C and 4.4 °C and evaluated for anthocyanin, flavonol, and chlorogenic acid content and percent polymeric color over eight weeks of storage. An ice pop product stored at −20 °C was evaluated for its anthocyanin and chlorogenic acid content over eight weeks of storage.

## 2. Materials and Methods

### 2.1. Standards and Solvents

HiActive^®^ North American wild blueberry powder was purchased from FutureCeuticals, Inc. (Momence, IL, USA). Rutin, chlorogenic acid, high performance liquid chromatography (HPLC) grade methanol, HPLC grade acetonitrile, potassium metabisulfite, formic acid, acetic acid, chlorogenic acid and rutin were purchased from Sigma-Aldrich (St. Louis, MO, USA). A standard mixture of delphinidin, cyanidin, petunidin, peonidin, pelargonidin, and malvidin glucosides was purchased from Polyphenols (Sandnes, Norway). 

### 2.2. Preparation of Blueberry Products

Samples of juice, ice pop, gummy, oatmeal bar, and graham cracker cookie, each containing 15 g of WBB powder per serving, were prepared and packaged as previously described [22]. One serving of oatmeal bar, ice pop, and graham cracker cookie was equivalent to one piece each (61 g, 121 g and 50 g, respectively), a juice serving was 135 g, and a gummy serving was 7 pieces, or 113 g. The amount of 15 g of WBB powder used in product formulations was calculated and converted from previous animal studies to humans [31]. The graham cracker cookies and oatmeal bars were prepared with minimal thermal treatment. This involved only the use of brief microwave heating to solubilize the ingredients in order to avoid thermal loss of phenolic compounds, but still obtain a ready-to-consume non-baked product. The blueberry juice and ice pop were prepared with an anthocyanin concentrate, previously extracted from the WBB powder [22]. This procedure was used to produce juice and ice pop products with no particulates. The formulation was adjusted with water so the anthocyanin content of the products was equivalent to that found in 15 g of WBB powder per serving. The preparation and processing of the samples for the storage study were performed in two separate experiments, using the same sample of wild freeze-dried blueberries obtained from FutureCeuticals Inc. (Momence, IL, USA). The WBB powder was stored at 15.5 °C for four months between the two experiments. The samples from Experiment 1 were stored at 21 °C and the samples from Experiment 2 were stored at 4.4 °C. The ice pop products prepared in Experiment 1 were stored at −20 °C. Three samples of each packaged product were evaluated at time 0 (immediately after preparation) and after 2, 4, 6, and 8 weeks of storage.

### 2.3. Extraction of Polyphenolics from Freeze-dried Blueberries and Products

Polyphenolics were extracted by homogenizing 5 g of WBB-containing food product or 1 g of WBB powder in 25 mL of extraction solution containing methanol/water/formic acid (60:37:3 v/v/v), to the smallest particle size using a Euro Turrax T18 Tissuemizer (Tekmar-Dohrman Corp, Mason, OH, USA) for 1 min. Homogenates were centrifuged for 5 min at 10,864 × *g*. The pellet was re-extracted two additional times with 25 mL of extraction solution and centrifuged for 5 min at 10,864 × *g*. The filtrates were pooled and adjusted to 100 mL with extraction solvent in a volumetric flask. Prior to HPLC analysis, 5 mL of extract were dried in a Thermo Savant Speed Vac Plus SC210A (Thermo Fisher Scientific, Waltham, MA, USA) and reconstituted in 1 mL 5% formic acid in water. All samples were passed through 0.45 µm nylon syringe filters (VWR, Radnor, PA, USA) into 1 mL HPLC vials prior to HPLC analysis. The ice pop and juice samples did not undergo extraction due to prior extraction of anthocyanins to make the concentrate used in the formulation but were filtered using the 0.45 µm nylon syringe filters prior to HPLC analysis.

### 2.4. HPLC Analysis of Anthocyanins and Chlorogenic Acid

Anthocyanins and chlorogenic acid were analyzed by HPLC using the method of Cho and others (2004) [7]. Samples (50 µL) were analyzed using a Waters HPLC system (Waters Corp, Milford, MA, USA) equipped with a model 600 pump, a model 717 Plus autosampler, and a model 996 photodiode array detector. Separation was carried out at room temperature using a 4.6 mm × 250 mm Symmetry C_18_ column (Waters Corp, Milford, MA, USA) preceded by a 3.9 mm × 20 mm Symmetry C_18_ guard column. The mobile phase was a linear gradient of 5% formic acid (A) and methanol (B) from 2% B to 60% B for 60 min at a flow rate of 1 mL/min. The system was equilibrated for 20 min at the initial gradient prior to each injection. Detection wavelengths of 320 nm and 510 nm were used to monitor chlorogenic acid and anthocyanin peaks, respectively. Individual anthocyanin monoglucosides and acylated anthocyanin derivatives were quantified as delphinidin, cyanidin, petunidin, peonidin, and malvidin glucoside equivalents using external calibration curves (3.75, 7.5, 15, 30, 60, 120, 240 mg/L; R^2^ > 0.9977 for each anthocyanin glucoside) of a mixture of authentic standards (Polyphenols, Sandnes, Norway). Chlorogenic acid was quantified using external calibration curves (4, 8, 16, 32, 64, 128, 256 mg/L; R^2^ = 0.9988) of an authentic standard (Sigma-Aldrich, St. Louis, MO, USA). Results are expressed as mg of anthocyanin or chlorogenic acid per g of WBB powder.

### 2.5. HPLC Analysis of Flavonols

Flavonols were analyzed by HPLC using the same HPLC system described above according to the method of Cho et al. (2005) [32]. Separation was performed at room temperature on a 4.6 mm × 250 mm Aqua C_18_ column (Phenomenex, Torrance, CA, USA) preceded by a 3.0 mm × 4.0 mm ODS C_18_ guard column. The mobile phase was a linear gradient of 2% acetic acid (A) and 0.5% acetic acid in water and acetonitrile (50:50 v/v) (B) from 10% B to 55% B in 50 min and from 55% B to 100% B in 10 min at a flow rate of 1 mL/min. The system was equilibrated for 20 min at the initial gradient prior to each injection. A detection wavelength of 360 nm was used to monitor flavonol peaks. Flavonols were quantified as rutin equivalents using an external calibration curve (3.3, 6.6, 13.2, 26.4, 52.8, 105.6, 211.2 mg/L; R^2^ = 0.9999) of an authentic standard (Sigma-Aldrich, St. Louis, MO, USA), with results expressed as mg of rutin equivalents per g of WBB powder.

### 2.6. HPLC/ESI-MS Analysis of Polyphenolics

An analytical Hewlett Packard 1100 series HPLC instrument (Palo Alto, CA, USA) equipped with an autosampler, binary HPLC pump, and UV/Vis detector was used. For HPLC/MS analysis, the HPLC apparatus was interfaced to a Bruker model Esquire-LC/MS ion trap mass spectrometer (Billerica, MA, USA). Mass spectral data were collected with the Bruker software (Bruker Co., DataAnalysis version 4.0, Billerica, MA, USA), which also controlled the instrument and collected the signal at 520 nm. Typical conditions for mass spectral analysis conducted in positive-ion electrospray mode for anthocyanins and negative-ion electrospray mode for flavonols included a capillary voltage of 4000 V, a nebulizing pressure of 30.0 psi, a drying gas flow of 9.0 mL/min, and a temperature of 300 °C. Data were collected in full scan mode over a mass range of *m/z* 50−1000 at 1.0 s per cycle. Characteristic ions (*m/z*) were used for peak assignment. For compounds where chemical standards were commercially available, retention times were also used to confirm the identification of components.

### 2.7. Analysis of Percent Polymeric Color

Percent polymeric color (% PC) of extracts was determined using the spectrophotometric assay of Giusti and Wrolstad (2001) [33]. Sample extracts were diluted with water in order to have an absorbance reading between 0.5 and 1.0 at 512 nm when evaluated by an 8452A Diode Array Spectrophotometer (Hewlett Packard, Palo Alto, CA, USA). For analysis, 0.2 mL of 0.90 M potassium metabisulfite was added to 2.8 mL diluted sample (bisulfite bleached sample) and 0.2 mL of DI water was added to 2.8 mL diluted sample (non-bleached, control sample). After equilibrating for 15 min, but not more than 1 h, samples were evaluated at λ = 700 nm, 512 nm, and 420 nm. Color density was calculated using the control sample according to the following formula:Color Density = [(A_420 nm_ − A_700 nm_) + (A_512 nm_ − A_700 nm_)] × Dilution Factor(1)

Polymeric color was determined using the bisulfite-bleached sample using the following formula:Polymeric Color = [(A_420 nm_ − A_700 nm_) + (A_512 nm_ − A_700 nm_)] × Dilution Factor(2)

Percent polymeric color was calculated using the formula:% Polymeric color = (polymeric color/color density) × 100(3)

### 2.8. Statistical Analysis

The effect of storage time (0, 2, 4, 6, and 8 weeks) on anthocyanins, flavonols, chlorogenic acid, and % polymeric color in each blueberry product was evaluated using the Fit Model platform of JMP (JMP Pro, version 15, SAS Institute, Cary, NC, USA)), and the percent retention of each compound after 8 weeks of storage was calculated using the fit model equation. The effect of storage temperature on phenolic compounds stability was not evaluated in this study due to the length of time (4 months) the WBB powder was stored between processing the products in Experiment 1 (products stored at 21 °C) and Experiment 2 (products stored at 4.4 °C). During this four-month storage time, the powder stored at 15.5 °C presumably absorbed moisture evident by powder clumping, resulting in different amounts of polyphenolics in the products immediately after processing. Principal component analysis (PCA) was performed with the total and individual anthocyanins variables, using the Multivariate platform in JMP, on the mean value (*n* = 3) of each sample per time point and using the correlation method. Correlations among total anthocyanins and percent polymeric color were determined by pairwise correlations method in the multivariate platform of JMP.

## 3. Results

### 3.1. Identification of Anthocyanins by HPLC/MS

The WBB powder used to prepare the products contained at least 22 anthocyanins (Table 1), which were identified by comparing their mass-to-charge (*m/z*) values and elution orders with previous studies [4,5,34]. Blueberries are unique in that three different sugars (galactose, glucose, arabinose) are commonly attached to the five anthocyanidins (delphinidin, cyanidin, petunidin, peonidin, malvidin) [34]. This was confirmed in our study; however, we were unable to detect peonidin-3-arabinoside using our HPLC method. We were unable to obtain complete separation of all of the anthocyanins present in the extract due to the complexity of the anthocyanin profile. Peak 15 contained two co-eluting compounds, namely cyanidin-3-(6″-malonyl) galactoside and cyanidin-3-(6″-acetyl) galactoside, and peak 18 was composed of three co-eluting compounds, namely delphinidin-3-rutinoside, cyanidin-3-(6″-malonyl) glucoside, and malvidin-3-(6″-acetyl) galactoside. We were unable to identify peak 17, which appeared to be a delphinidin derivative based on its aglycone *m/z* of 303, but the molecular ion *m/z* value was ambiguous. Many of the anthocyanins were present in acylated form. Two of the cyanidin glycosides (galactoside and glucoside) were acylated with malonic acid, whereas delphinidin, cyanidin, and malvidin galactosides as well as petunidin, peonidin, and malvidin glucosides were acylated with acetic acid moieties.

### 3.2. Identification of Flavonols by HPLC/MS

The WBB powder used to prepare the products contained at least 12 flavonols (Table 2), which were identified by comparing their mass-to-charge (*m/z*) values and elution orders with previous studies [9,32]. The WBB powder contained one syringetin derivative, three myricetin derivatives, and eight quercetin derivatives. We were unable to identify peak 8, which appeared to be a quercetin derivative based on its aglycone *m/z* of 300, but the molecular ion *m/z* of 623 was ambiguous.

### 3.3. Stability of Anthocyanins in Blueberry Products during Storage

#### 3.3.1. Ice Pop

The total anthocyanin content of the ice pop over eight weeks of storage at −20 °C is shown in Figure 1. The total amount of anthocyanins significantly decreased with storage time (*p* = 0.0056), but the percent retention remained high with 93% of total anthocyanins retained in the product after eight weeks. Consistent with our results, total anthocyanin content of frozen blueberries was stable over three months of storage at −20 °C [35]. Changes in major individual anthocyanins in the ice pop over eight weeks of storage at −20 °C are shown in Appendix A. Most of the individual anthocyanins did not significantly decrease over storage. For the anthocyanins that decreased during storage, their percent retention after eight weeks remained over 87%: malvidin-3-glucoside (87.3%), malvidin-3-galactoside (94.5%), cyanidin-3-galactoside (91.2%), malvidin-3-(6″-acetyl) glucoside (91.1%), petunidin-3-glucoside (91.4%).

#### 3.3.2. Oatmeal Bar

The total anthocyanin content of the oatmeal bar decreased with storage time at 4.4 °C (*p* = 0.0008) and 21 °C (*p* = 0.0008). The mean total anthocyanin content of both storage temperatures over eight weeks of storage is shown in Figure 2. After eight weeks of storage, the oatmeal bar retained 86.4% of the total anthocyanins present in the control sample (Day 0) when stored at 4.4 °C and 74.1% when stored at 21 °C. Changes in the major individual anthocyanins in the oatmeal bar over eight weeks of storage are shown in Appendix A. At 4.4 °C, seven individual anthocyanins did not significantly decrease over storage: petunidin-3-(6″-acetyl) glucoside, petunidin-3-arabinoside, petunidin-3-glucoside, delphinidin-3-(6″-acetyl) glucoside, cyanidin-3-arabinoside, the two co-eluting compounds (cyanidin-3-(6″-malonyl) galactoside + cyanidin-3-(6″-acetyl) galactoside), and the three co-eluting compounds (delphinidin-3-rutinoside + malvidin-3-(6″-acetyl) galactoside + cyanidin-3-(6″-malonyl) glucoside). The percent retention values of the anthocyanins over eight weeks of storage at 4.4 °C were all > 75%. When stored at 21 °C, petunidin-3-arabinoside, malvidin-3-(6″-acetyl) glucoside, the unknown delphinidin derivative, and the two co-eluting compounds (cyanidin-3-(6″-malonyl) galactoside + cyanidin-3-(6″-acetyl) galactoside) did not significantly decrease over eight weeks of storage. The percent retention of the other anthocyanins ranged from 68.2% (malvidin-3-glucoside) to 82.8% (petunidin-3-galactoside).

#### 3.3.3. Graham Cracker Cookie

The total anthocyanin content of the graham cracker cookie decreased with storage time at 4.4 °C (*p* = 0.0003) and 21 °C (*p* < 0.0001). The mean total anthocyanin content of both storage temperatures over eight weeks of storage is shown in Figure 2. After eight weeks of storage, the graham cracker cookie retained about 74% of the total anthocyanins present in the control sample (Day 0) for both storage temperatures. Changes in the major individual anthocyanins in the graham cracker cookie over eight weeks of storage are shown in Appendix A. At both storage temperatures, petunidin-3-arabinoside and the two co-eluting compounds (cyanidin-3-(6″-malonyl) galactoside + cyanidin-3-(6″-acetyl) galactoside) did not significantly decrease with storage time. Percent retention of the three delphinidin-3-glycosides (glucoside, galactoside, and arabinoside) was >80% at 4.4 °C. However, the unidentified delphinidin derivative showed only 32.9% retention, while all the other compounds showed retentions >50%. When stored at 21 °C, delphinidin-3-arabinoside and petunidin-3-galactoside showed >80% retentions, while the unknown delphinidin derivative was the only compound showing <50% retention after eight weeks of storage (47.1%).

#### 3.3.4. Juice

The total anthocyanin content of the juice decreased with storage time for each storage temperature (*p* < 0.0001). The total anthocyanin content of juice stored at 4.4 °C and 21 °C is shown in Figure 2. After eight weeks of storage, the juice stored at 4.4 °C retained 90.7% of total anthocyanins compared with control samples (Day 0), whereas the juice stored at 21 °C retained 69.1%. Concentrations of anthocyanins are known to readily decline during storage of blueberry juice at ambient temperature [23,36], but refrigeration is an effective treatment to ameliorate anthocyanin losses [29,37,38]. Changes in the major individual anthocyanins in the juice stored at 4.4 °C and 21 °C over eight weeks of storage are shown in Appendix A. At 4.4 °C, peonidin-3-galactoside, cyanidin-3-arabinoside, malvidin-3-galactoside, malvidin-3-glucoside, and malvidin-3-(6″-acetyl) galactoside remained stable over the eight weeks of storage. At 4.4 °C, all anthocyanins showed >50% retention, with the minimal percent retention being 57.7% for the unknown delphinidin derivative. This compound, however, did not significantly decrease over storage at 21 °C, along with the two co-eluting anthocyanins (cyanidin-3-(6″-malonyl) galactoside + cyanidin-3-(6″-acetyl) galactoside). Besides these two compounds, the percent retention of anthocyanins at 21 °C ranged from 59% (malvidin-3-(6″-acetyl) glucoside) to 75.5% (petunidin-3-glucoside).

#### 3.3.5. Gummy Product

The total anthocyanin content of the gummy product decreased with storage time for each storage temperature (*p* < 0.0001). The total anthocyanin content of the gummy product stored at 4.4 °C and 21 °C is shown in Figure 2. After eight weeks of storage, the gummy product stored at 4.4 °C and 21 °C retained 43.2% and 50.6%, respectively, of their original total anthocyanin content (Day 0). Consistent with our findings, levels of total anthocyanins declined in gelatin gels prepared with grape pomace extract over 24 weeks of storage at 21 °C, with losses most pronounced in gels exposed to neon light [39]. Maier et al. (2009) [39] also reported similar retention of total anthocyanins in gels stored for 24 weeks at 6 °C and 24 °C. The lower amount of anthocyanins recovered in the gummies stored at 4.4 °C compared to the same product stored at 21 °C may be explained by reduced extraction efficiency due to the hardening of the gel at low temperature, as opposed to degradation late during storage. Changes in the major individual anthocyanins in the gummy product stored at 4.4 °C and 21 °C over eight weeks of storage are shown in Appendix A. At 4.4 °C, all the individual anthocyanins decreased with storage time with retentions <50%, except for the two co-eluting anthocyanins (cyanidin-3-(6″-malonyl) galactoside + cyanidin-3-(6″-acetyl) galactoside) (60.9%) and malvidin-3-glucoside (52.1%). The percent retentions of the rest of the anthocyanins at this storage temperature ranged from 29.3% to 49.2%. When stored at 21 °C, two anthocyanins did not significantly decrease with time, namely the unknown delphinidin derivative and the two co-eluting compounds (cyanidin-3-(6″-malonyl) galactoside + cyanidin-3-(6″-acetyl) galactoside). For the rest of the anthocyanins, percent retentions ranged from 40% (cyanidin-3-glucoside and cyanidin-3-galactoside) to 71% (malvidin-3-(6″-acetyl) glucoside).

In all the products, the individual anthocyanin loss did not appear to be impacted by the anthocyanidin structure or the type of sugar moiety attached (data not shown).

#### 3.3.6. Product Comparison of Anthocyanin Composition over Eight Weeks of Storage

The distribution of the products according to their individual anthocyanin profile as affected by storage time (0, 2, 4, 6, and 8 weeks) can be visualized on a PCA scores plot (Figure 3). The first principal component (PC1) explained 83.9% of the variation with all the individual anthocyanins being positively loaded on PC1. Therefore, PC1 represents the amount of individual anthocyanins. The juice and ice pop samples had high scores on PC1 (except for the juice samples stored at 21 °C for 6 and 8 weeks). The oatmeal bar samples also had positive scores on PC1 for the earlier storage times, whereas the oatmeal bar samples stored at 21 °C for eight weeks were the only oatmeal sample to have a negative score. Except for the control samples (Day 0), all the graham cracker cookie samples had negative scores on PC1, regardless of the storage temperature. Finally, all gummy samples (for both storage at 4.4 °C and 21 °C) had negative scores on PC1, with scores becoming smaller with storage time. The PCA figure confirmed higher values of anthocyanins in the juice and ice pop samples, as well as, to a lesser extent, the oatmeal bars. The graham cracker cookie and gummy samples did not demonstrate high values for anthocyanins, with a clear loss of anthocyanins with storage time for the gummy samples.

### 3.4. Changes in Percent Polymeric Color in Blueberry Products during Storage

Changes in percent polymeric color (% PC) values in oatmeal bar, graham cracker cookie, juice, and gummy product stored at 4.4 °C and 21 °C over eight weeks of storage are shown in Figure 4.

#### 3.4.1. Oatmeal Bar

Percent polymeric color values of oatmeal bar samples increased over storage at 4.4 °C (*p* = 0.0012) and 21 °C (*p* = 0.0173). For samples stored at 4.4 °C, % PC values increased from 7% at day 0 to 9.6% at eight weeks, and from 5.7% at day 0 to 9.6% at eight weeks in samples stored at 21 °C. Percent polymeric color values showed significant inverse correlations to levels of total anthocyanins at 4.4 °C (r_xy_ = −0.65; *p* = 0.0081) and at 21 °C (r_xy_ = −0.64; *p* = 0.0103) during storage.

#### 3.4.2. Graham Cracker Cookie

Percent polymeric color values of graham cracker cookie samples significantly increased over storage at 4.4 °C (*p* < 0.0001), but not at 21 °C (*p* = 0.2075). For samples stored at 4.4 °C, % PC values increased from 2.5% at day 0 to 6.3% at eight weeks, but when stored at 21 °C, % PC values remained at 6.2% on average during eight weeks of storage. Percent polymeric color values showed significant inverse correlations to levels of total anthocyanins at 4.4 °C (r_xy_ = −0.56; *p* = 0.0298) and 21 °C (r_xy_ = −0.58; *p* = 0.0237) during storage.

#### 3.4.3. Juice

Percent polymeric color values of juice samples significantly increased over storage at 4.4 °C (*p* = 0.0008) and 21 °C (*p* < 0.0001). For samples stored at 4.4 °C, % PC values increased from 13.6% at day 0 to 20.8% at eight weeks, and the values increased from 13.8% at day 0 to 23.5% at eight weeks when stored at 21 °C. Percent polymeric color values showed significant inverse correlations to levels of total anthocyanins at 4.4 °C (r_xy_ = −0.58; *p* = 0.0239), and more strongly at 21 °C (r_xy_ = −0.71; *p* = 0.0029) during storage.

#### 3.4.4. Gummy Product

Percent polymeric color values of gummy samples significantly increased over storage at 4.4 °C (*p* = 0.0229) and 21 °C (*p* < 0.0001). For samples stored at 4.4 °C, % PC values increased from 6.8% at day 0 to 12.7% at eight weeks, and in samples stored at 21 °C, % PC values increased from 6.9% at day 0 to 23.8% at eight weeks of storage. Percent polymeric color values showed a moderate inverse correlation to levels of total anthocyanins at 4.4 °C (r_xy_ = −0.563; P = 0.036), but a much greater inverse correlation at 21 °C (r_xy_ = −0.93; *p* < 0.0001) during storage.

Percent polymeric color values typically show an inverse correlation with total anthocyanins during storage of blueberry products [23,24], and inverse correlations with each individual anthocyanins in all the products and storage temperature (data not shown). Higher percent polymeric color values indicate that a higher percentage of anthocyanins are resistant to bleaching in the presence of potassium metabisulfite. Since the sulfonic acid adduct attaches at C4 on the middle heterocyclic ring, it is thought that anthocyanin–procyanidin polymers are formed via a direct condensation reaction, resulting in a C4–C8 anthocyanin–procyanidin linkage as the major polymers formed in blueberries during storage. Hence, it is possible that declines in anthocyanins during storage of the blueberry products are not true losses due to degradation, but the conversion of monomeric anthocyanins to anthocyanin–procyanidin polymers. Anthocyanins can be degraded via a hydration reaction, where the flavylium ion is converted to a hemiketal structure, which is rapidly converted to *cis*-chalcone, which slowly arranges to a *trans*-chalcone structure [40]. The *trans*-chalcone structure is highly unstable and rapidly degrades to hydroxybenzoic acid derivatives [40]. However, we do not consider that this reaction was responsible for anthocyanin losses in the blueberry products over storage since we did not observe an increase in phenolic acid derivatives in our HPLC chromatograms at 280 nm (data not shown).

### 3.5. Stability of Chlorogenic Acid during Storage

The stability of chlorogenic acid in the four blueberry products stored at 4.4 °C and 21 °C is shown in Figure 5. Chlorogenic acid was stable in all products over storage regardless of storage temperature, except for the juice and oatmeal bar stored at 4.4 °C, where levels significantly decreased (*p* = 0.0249 and *p* = 0.0133, respectively). At 4.4 °C, the chlorogenic acid content decreased from 4.3 to 3.6 mg/g WBB powder in the juice and from 3.0 to 2.6 mg/g WBB powder in the oatmeal bar. At 21 °C, chlorogenic acid in the juice showed a slight increasing trend; however, this change was not statistically significant (*p* = 0.6496). Chlorogenic acid was also stable in the ice pop over eight weeks of storage at −20 °C (*p* = 0.8332), with an average value of 6.5 mg/g WBB powder over storage (Figure 1). Initial levels of chlorogenic acid were higher in all products stored at 21 °C compared with 4.4 °C storage, which may be due to the variation in processing the two sets of samples for the storage study, or possible the degradation of chlorogenic acid in the WBB powder used to prepare the products. The WBB powder used to prepare samples for the refrigerated storage study was stored at 15.5 °C for three months prior to preparing the samples. Blueberries contain polyphenol oxidase, which can readily oxidize chlorogenic acid [41]. Chlorogenic acid was previously found to be stable in blueberry juice, puree, and canned berries stored for six months at 25 °C [23], but blueberry jams lost 27% of chlorogenic acid over six months of storage at 25 °C [24].

### 3.6. Stability of Flavonols in Blueberry Products during Storage

The stability of total flavonols in the five blueberry products is shown in Figure 6. Total flavonol levels in the oatmeal bar stored at both temperatures and in the gummy product and graham cracker cookie stored at 4.4 °C were stable over eight weeks of storage as well as the juice samples stored at 4.4 °C from two to eight weeks (*p* > 0.05). Flavonols in the juice stored at 21°C were also stable over time despite an upward trend, but the increase was not significant (*p* = 0.1935). Consistent with our findings, total flavonol concentrations were found to be relatively stable (<15% losses) in blueberry jam, juice, puree, and canned berries over six months of storage at 25 °C [24,42].

However, total flavonol content significantly decreased over storage in the gummy product (*p* = 0.0085) and graham cracker cookie (*p* = 0.0237) stored at 21 °C. Total flavonol levels declined by 45.7% and 28.5%, respectively, in these products over eight weeks of storage. In the gummy product, the most marked loss occurred from six to eight weeks of storage. Moisture loss during late storage presumably led to hardening of the gummies, resulting in incomplete extraction of the flavonols.

Individual flavonols in oatmeal bar and juice were all stable for both storage temperature as well as in graham cracker cookie stored at 4.4 °C (data not shown). In the graham cracker cookie stored at 21 °C, myricetin-3-galactoside was the most unstable flavonol showing 57.8% retention after eight weeks (*p* = 0.038). Quercetin-3-rutinoside (*p* = 0.0225), quercetin-3-galactoside (*p* = 0.0195), quercetin-3-glucoside (*p* = 0.037), quercetin-3-rhamnoside (*p* = 0.0058), quercetin-3-(6″-acetyl) galactoside (*p* = 0.0195), and syringetin-3-galactoside/glucoside (*p* = 0.0237) were all retained at levels from 61.1% to 70.1%. Four individual flavonols significantly decreased in the gummy product stored at 4.4 °C. The percent retention of quercetin-3-glucuronide (*p* = 0.0012) and quercetin-3-rhamnoside (*p* = 0.0016) were 33% and 38.5%, respectively. In the gummy product stored at 21 °C, only two flavonols (myricetin-3-glucoside and the syringetin derivative) showed no significant decrease over eight weeks of storage. All other flavonols in the gummy product stored at 21 °C were adversely affected by storage time with retentions ranging between 45.2 and 60.1% after eight weeks.

## 4. Conclusions

The stability of polyphenolics over eight weeks of storage in food products made with WBB powder varied according to product type. Polyphenolic compounds from the ice pop, oatmeal bar, and juice were shown to be stable over storage and are good candidates for further use in applications in which stored food items are to be used for delivery of significant amounts of polyphenolics (e.g., controlled feeding trials). The other food items may also be used for these applications, but the relative retention of bioactive polyphenolics as outlined herein should be taken into account when dose and delivery are designed. The gummy product showed relatively poor retention of anthocyanins and flavonols after eight weeks of storage, which may be due to extraction issues rather than true losses. In summary, incorporating WBB powder into food products in which key molecules remain intact during storage can improve the consumption of blueberry phytochemicals. The effect of the product matrix on bioavailability of retained polyphenolic compounds needs further investigation.

## Figures and Tables

**Figure 1 foods-09-00466-f001:**
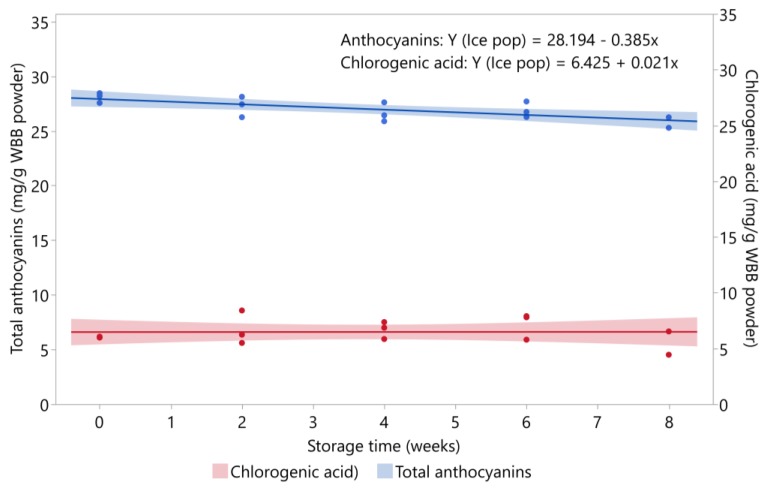
Stability of total anthocyanins and chlorogenic acid in ice pop stored at −20 °C (*n* = 3/time point). Shaded area around lines represents 95% confidence intervals for predicted values.

**Figure 2 foods-09-00466-f002:**
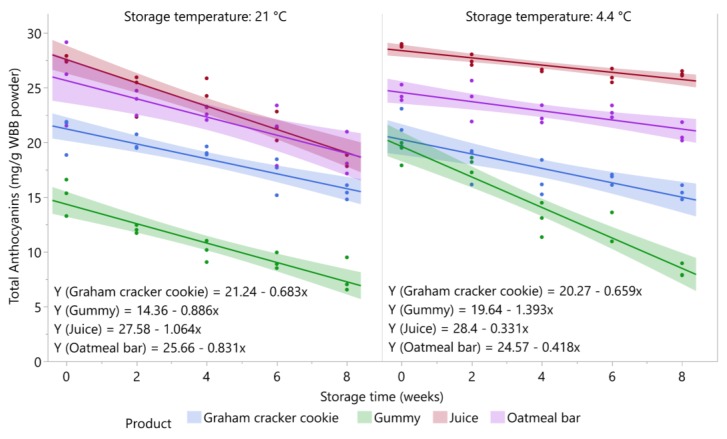
Stability of total anthocyanins in blueberry stored at 21 °C and 4.4 °C (*n* = 3/time point). Shaded area around lines represents 95% confidence intervals for predicted values.

**Figure 3 foods-09-00466-f003:**
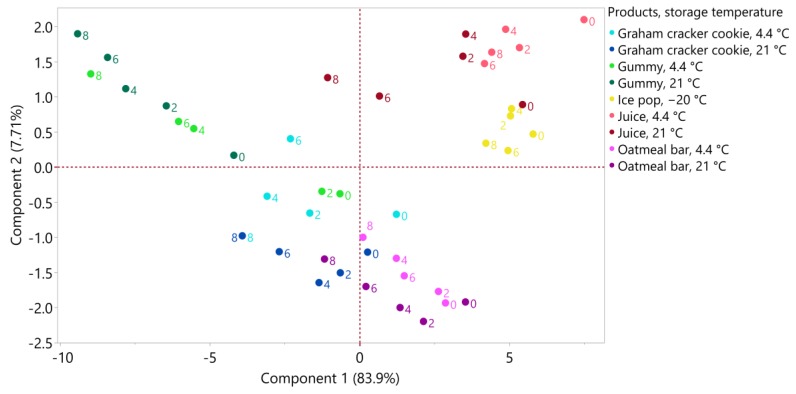
Principal component analysis (PCA) scores plot of blueberry products stored at −20 °C (ice pop), 4.4 °C and 21 °C (graham cracker cookie, gummy, juice, oatmeal bar) for 0, 2, 4, 6, and 8 weeks (*n* = 3/time point).

**Figure 4 foods-09-00466-f004:**
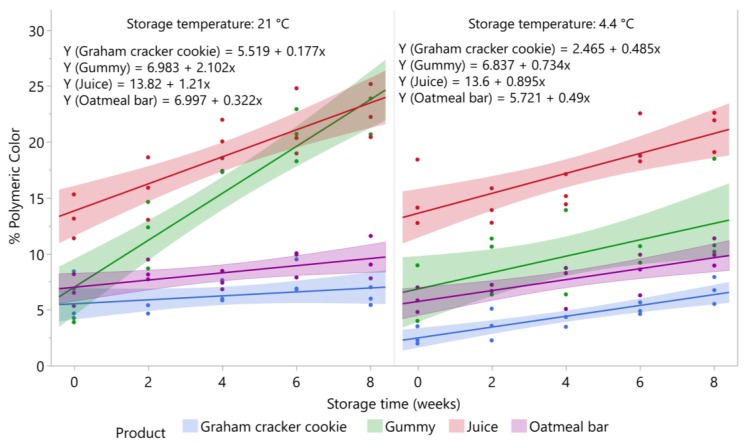
Stability of percent polymeric color in blueberry stored at 21 °C and 4.4 °C (*n* = 3/time point). Shaded area around lines represents 95% confidence intervals for predicted values.

**Figure 5 foods-09-00466-f005:**
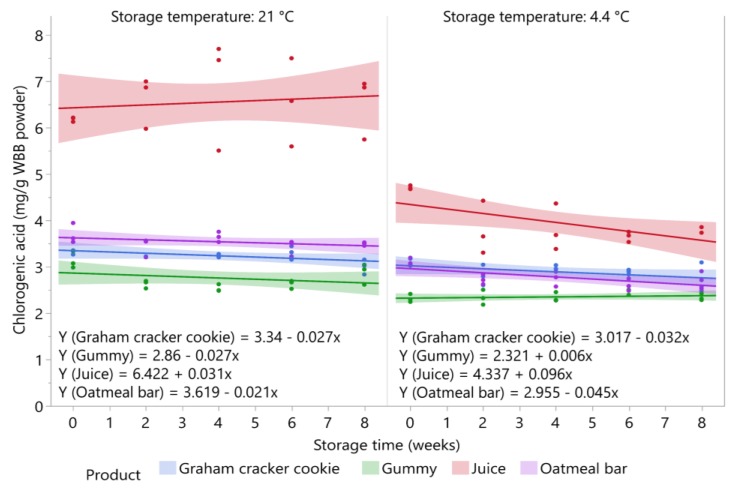
Stability of chlorogenic acid in blueberry stored at 21 °C and 4.4 °C (*n* = 3/time point). Shaded area around lines represents 95% confidence intervals for predicted values.

**Figure 6 foods-09-00466-f006:**
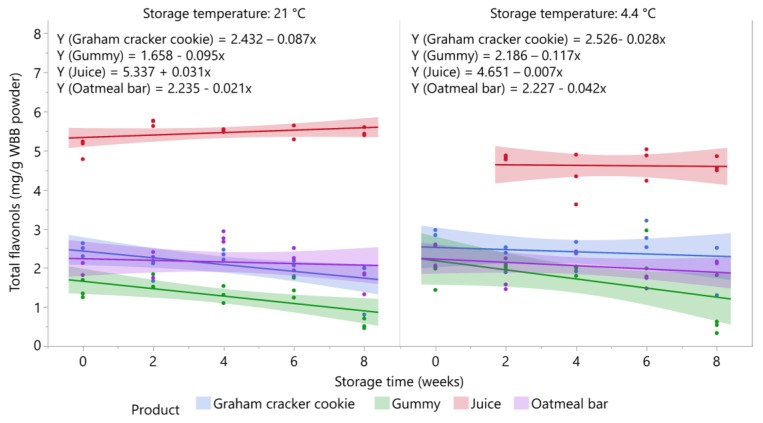
Stability of total flavonols in blueberry stored at 21 °C and 4.4 °C (*n* = 3/time point). Shaded area around lines represents 95% confidence intervals for predicted values.

**Table 1 foods-09-00466-t001:** Peak assignments, retention times (RT in min), and mass spectral data of anthocyanins in extract from WBB powder.

Peak	HPLC (RT in min)	Identification	*m/z* [M^+^]	Fragments
1	24.9	delphinidin-3-galactoside	465	303
2	26.2	delphinidin-3-glucoside	465	303
3	27.1	cyanidin-3-galactoside	449	287
4	28.1	delphinidin-3-arabinoside	435	303
5	28.8	cyanidin-3-glucoside	449	287
6	29.9	petunidin-3-galactoside	479	317
7	30.3	cyanidin-3-arabinoside	419	287
8	31.1	petunidin-3-glucoside	479	317
9	31.8	peonidin-3-galactoside	463	301
10	32.9	petunidin-3-arabinoside	449	317
11	33.2	peonidin-3-glucoside	463	301
12	33.6	malvidin-3-galactoside	493	331
13	35.1	malvidin-3-glucoside	493	331
14	36.6	malvidin-3-arabinoside	463	331
15	37.7	cyanidin-3-(6″-malonyl) galactoside + cyanidin-3-(6″-acetyl) galactoside	535 491	287 287
16	40.0	delphinidin-3-(6″-acetyl) galactoside	507	303
17	41.6	delphinidin derivative	-	303
18	42.3	delphinidin-3-rutinoside malvidin-3-(6″-acetyl) galactoside cyanidin-3-(6″-malonyl) glucoside	611 535 535	303 331 287
19	42.7	malvidin-3-(6″-acetyl) galactoside	535	331
20	43.6	petunidin-3-(6″-acetyl) glucoside	521	317
21	45.7	peonidin-3-(6″-acetyl) glucoside	505	301
22	46.6	malvidin-3-(6″-acetyl) glucoside	535	331

**Table 2 foods-09-00466-t002:** Peak assignments, retention times (RT in min), and mass spectral data of flavonols in extract from WBB powder.

Peak	HPLC (RT in min)	Identification	*m/z* [M^−^]	Fragments
1	39.2	myricetin-3-galactoside	479	316
2	39.8	myricetin-3-glucoside	479	316
3	43.8	myricetin-3-rhamnoside	463	316
4	44.2	quercetin-3-rutinoside	609	300
5	45.0	quercetin-3-galactoside	463	300
6	45.8	quercetin-3-glucoside	463	300
7	47.0	quercetin-3-glucuronide	477	301
8	48.9	Unknown	623	505, 433, 300
9	50.3	quercetin-3-pentoside	433	300
10	51.0	quercetin-3-rhamnoside	447	300
11	51.1	syringetin-3-galactoside/glucoside	507	344
12	51.8	quercetin-3-(6″-acetyl) galactoside	505	300

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
