# Peer review of "Changes in Polyphenolics during Storage of Products Prepared with Freeze-Dried Wild Blueberry Powder"

_foods, 2020, doi:10.3390/foods9040466_

Round 1

Reviewer 1 Report

The paper is well-written, scientifically sound, presented in a proper style.

Some minor remarks:

  • The appropiate names for the acyl groups should be given (lines 32-33).
  • The authors generally give plausible explanations for all the changes observed. However, they did not interpret the observed increase in chlorogenic acid in the case of fruit juice at room temperature, being apparently in contradiction with the decreasing tendency observed at 4 °C. The same applies for total flavonols.

Reviewer 2 Report

The paper is interesting, I would suggest revising the following items:

Line 21. It's a little bit surprising that the losses of anthocyanins are higher at 4.4 ºC. This fact is commented again in line 264. Some references are given reinforcing the obtained results at room temperature, and details are presented about losses/retention, but an explanation and discussion about them should be given.

Line 79. Are the non-baked products comparable to the real, finished foods? It is clear that the thermal process will decrease the content of phenolic compounds, and other chemical species can be formed. Nevertheless, working with final products would be more interesting from a real use point of view.

Line 80. The concentrate, prepared with ethanol extraction according to ref. [22] has a different concentration and profile of anthocyanins which surely have influence on final results.

Line 83. Please, specify the amount corresponding per serving in all the cases.

Line 86. What is the reason for the 4-month delay? Intentional or simply organization purposes?

Line 91. What is the criterion for the selection of 5 g of food product or 1 g of WBB powder? Limits of detection? Linear range? Please, specify.

Line 114. Please, give details about number of points, linear range, detection limits and R2 value.

Line 161. "may have absorbed". The moisture content has been determined before and after the storage time or "may" is only a hypothesis? This fact can be important in the determination of accurate masses as well as some reactions could occur or extraction efficiency could be influenced.

Figure 1. No comment at all is given about the evolution of chlorogenic acid. In addition, there is a point clearly different (lower) at 8 weeks. In other figures, there are numerous individual points clearly out of the colored band indicating 95% confidence intervals. The consideration of these values for calculations can seriously modify the calculated equations, thus changing the interpretation about stability/losses. Have authors considered a rejection criterion?

Figure 2. Same comment about the dispersion of points, mainly in Graham cracker cookies and gummy. In some cases, only one out of three points per time appears into the confidence interval.

Line 244. Please, revise numbering.

Lines 261-277. Please, provide a discussion to justify the just reported values.

Figure 5. There is a huge dispersion of values for juice samples. The results/conclusions obtained must be seriously considered, since their validity can be compromised.

Line 368. Please, provide p-values as in the previous sections.

Figure 6. Huge dispersion of values mainly for crackers and gummies (4 weeks at 21 ºC and 6&8 weeks at 4.4 ºC). Same comment than previous figures.

Line 399. The extraction recovery has been calculated. Probably a standard addition could be considered for determining matrix effects, thus discriminating the sources of the obtained results.

Supplementary figures. It is difficult to distinguish significant differences, mainly for lower traces. Besides, the high dispersion values of error bars in some cases could suggest lack of homogeneity among samples. As mentioned several times, authors should evaluate RSD values and consider rejection criteria for anomalous results. Then, a reconsideration of trends and a deeper explanation should be given.

Round 2

Reviewer 2 Report

Almost all the suggestions have been satisfactorily taken into account. It would be desirable higher precision as well as the moisture control, but it would require the repetition of a significant part of the experiments and a long time, which is no intended at all. So, there is no objection to accept the paper for publication.